# An Immunoinformatics Approach to Design Novel and Potent Multi-Epitope-Based Vaccine to Target Lumpy Skin Disease

**DOI:** 10.3390/biomedicines11020398

**Published:** 2023-01-29

**Authors:** Muhammad Shahab, A. Khuzaim Alzahrani, Xiuyuan Duan, Muneeba Aslam, Mohd. Imran, Mehnaz Kamal, Md. Tauquir Alam, Guojun Zheng

**Affiliations:** 1State Key Laboratories of Chemical Resources Engineering, Beijing University of Chemical Technology, Beijing 100029, China; 2Medical Laboratory Technology, Faculty of Applied Medical Sciences, Northern Border University, Arar 91431, Saudi Arabia; 3Department of Biochemistry, Abdul Wali Khan University, Mardan 23200, Pakistan; 4Department of Pharmaceutical Chemistry, Faculty of Pharmacy, Northern Border University, Rafha 91911, Saudi Arabia; 5Department of Pharmaceutical Chemistry, College of Pharmacy, Prince Sattam Bin Abdulaziz University, Al-Kharj 11942, Saudi Arabia

**Keywords:** LSD, multi-epitope vaccine, immunoinformatics approach, molecular docking, MD simulation

## Abstract

The lumpy skin disease (LSD) virus of the Poxviridae family is a serious threat that mostly affects cattle and causes significant economic loss. LSD has the potential to spread widely and its rapidly across borders. Despite the availability of information, there is still no competitive vaccine available for LSD. Therefore, the current study was conducted to develop an epitope-based LSD vaccine that is efficient, secure, and biocompatible and stimulates both innate and adaptive immune responses using immunoinformatics techniques. Initially, putative virion core proteins were manipulated; B-cell and T-cell epitopes have been predicted and connected with the help of adjuvants and linkers. Numerous bioinformatics methods, including antigenicity testing, transmembrane topology screening, allergenicity assessment, conservancy analysis, and toxicity evaluation, were employed to find superior epitopes. Based on promising vaccine candidates and immunogenic potential, the vaccine design was selected. Strong interactions between TLR4 and TLR9 and the anticipated vaccine design were revealed by molecular docking. Finally, based on the high docking score, computer simulations were performed in order to assess the stability, efficacy, and compactness of the constructed vaccine. The simulation outcomes showed that the polypeptide vaccine design was remarkably stable, with high expression, stability, immunogenic qualities, and considerable solubility. Additionally, computer-based research shows that the constructed vaccine provides adequate population coverage, making it a promising candidate for use in the design of vaccines against other viruses within the Poxviridae family and potentially other virus families as well. These outcomes suggest that the epitope-based vaccine developed in this study will be a significant candidate against LSD to control and prevent LSDV-related disorders if further investigated experimentally.

## 1. Introduction

Lumpy skin disease (LSD), which is associated with the Poxviridae family, is a serious threat to cattle stockbreeding. Lumpy skin disease is caused by the lumpy skin virus (LSDV), which can induce high fever and extensive nodules on the mucosa or the scarf skin in cattle. It is transmitted by blood-feeding insects, such as certain species of flies and mosquitoes, or ticks. Indeed, an efficient vaccine’s effect is its ability to provide life-long immunity, protecting against repeated episodes of infection. Around the globe, mainly in Asia, Europe, and the Middle East, the infection has become a deadly concern for large domesticated ruminants [1]. According to the World Organization for Animal Health (WOAH), because of the massive financial losses and the possibilities for spread, it has been identified and added to the list of notifiable diseases due to its potential for rapid cross-border spread [2,3]. Animals of all ages and breeds are impacted, although the young and those lactating at their peak are most vulnerable [4]. Because it causes a sharp decrease in milk production, abortions, poor coat health, and sterility in bulls, the disease is devastating [5]. It spreads so quickly across continents if infected animals are moved between farms and quarantine protocols are relaxed [6]. However, there is no epidemiological proof that the disease is contagious among animals [7]. The disease was confined to large parts of Africa until 1988, before gradually spreading to the Middle East, Eastern Europe, and ultimately the Russian Federation [8]. New cases were then recorded in South and East Asia in 2019 as the pandemic continued to expand [9,10]. The reappearance of the lumpy skin virus in different regions of the world highlighted the need to reconsider the nature of the illness, the viral transmission mechanism, and modernized preventative and adaptive control approaches. A comprehensive assessment of LSD has been done, primarily focusing on the south-eastern region of Asia, taking into account the aforementioned facts. LSD is a double-stranded DNA virus that is roughly 150 kb in size, is a member of the Capripoxvirus genus, and has genetic ties to the sheep pox and goat pox viruses [11]. The virus maintains a constant pH between 6.6 and 8.6 and is typically resistant to numerous physical and chemical agents, but it is inclined to more alkaline environments [12]. It only has the potential to survive for thirty-three days in necrotic skin nodules, thirty-five days in desiccated crusts, six months in sunlight-protected diseased tissue, and at least 18 days in air-dried hides at room temperature. The standard method of developing vaccines is costly and time-consuming. The multi-epitope vaccine method based on reverse vaccinology is nevertheless an attractive option. To combat human diseases caused by the monkeypox virus, a close family member of the variola virus, the causative agent of smallpox, which killed 300 million people worldwide in the twentieth century; Burkholderia pseudomallei [13]; SARS-CoV-2 [14], a global outbreak of a respiratory illness (coronavirus disease (COVID-19)) caused by a newly discovered coronavirus variant, SARS-CoV-2, which threatened human existence by claiming 54.76 lakhs lives as of 5 January 2022 [15] (the WHO designated Alpha, Beta, Gamma, Delta, and Omicron as SARS-CoV-2 variants of concern); Mokola Rabies virus [16]; and Congo virus, various vaccines are being created using reverse vaccinology techniques [17]. Herein, we developed an epitope-based lumpy skin disease vaccine that is efficient, secure, and biocompatible and stimulates both specific and non-specific immune responses. The present research aims to limit the abundance of lumpy skin disease and to achieve a suitable epitope vaccine component against LSDV. In this case, we used immunoinformatics to create the most antigenic and non-allergic vaccine epitopes. The discoveries lay the foundation for the development of a lumpy skin disease vaccine, but additional experimental verification is required to lower the burden of lumpy skin disease.

## 2. Methodology

The overall mechanism and various tools used in this study to design a multi-epitope vaccine by in silico processes are depicted in Figure 1.

### 2.1. Sequence Data Retrieval and Phylogenetic Analysis

Sequences were obtained from the publicly available NCBI database using the accession number (QIN91633.1). The non-redundant sequences were obtained via NCBI blast tools and were subjected to alignment through the MUSCLE program, and the top 11 sequences (along with the reference sequence) were collected using a FASTA file. Then, the non-redundant FASTA file was used for phylogenetic tree analysis, which was constructed using Mega-X. All of the obtained sequences were assessed for antigenicity and allergenicity using VaxiJen v2.0 (http://www.ddg-pharmfac.net/vaxijen/VaxiJen/VaxiJen.html) accessed on 25 November 2022 [18] and AllerTOP v2.0 (https://www.ddg-pharmfac.net/AllerTOP/ accessed on 25 November 2022) [19] server, respectively. The threshold parameter of the VaxiJen server was set to 0.4 throughout this study. The protein with the highest antigenicity and non-allergenic properties was selected for further analysis.

### 2.2. B- and T-Cell Lymphocyte Prediction 

Highly antigenic and non-allergic sequences were subjected to antibody recognition, and such sequences may be used to predict B-cell epitopes [20]. The smallest immunogenic peptides found in an antigen, known as the epitope, have the capacity to activate the immune responses [21]. The epitopes for immunological responses were predicted using the IEDB resource (http://tool.iedb.org/main/ accessed on 20 December 2022) [14]. T-cell epitope binding scores with MHC-I and MHC-II were computed. The final selection was chosen when the IC50 values were <100. The lower the IC50, the higher the binding affinity.

### 2.3. Development of Vaccine Based on Cytotoxicity, Allergenicity, and Antigenicity

The VaxiJen server performed the antigenicity assessment with an accuracy percentage between 70 and 89% [22]. The antigenic calling threshold was set at >0.5 VaxiJen probability score, as previously stated [23]. The allergenicity and toxicity of the epitopes were predicted using AllerTOP and ToxinPred, respectively. The construction of the multi-epitope vaccine was done using the mentioned standards. To create multi-epitope vaccination constructs, the lead epitopes were linked together using adjuvants and amino acid linkers (such as 50S ribosomal proteins, CPGPG, and AAY) [24]. The HLA-DR-binding epitope was also added to the adjuvant in order to improve the efficacy [25]. The constructions made up of various combinations were tested for low toxicity or allergenicity and high immunogenicity. AllerTOP anticipated the non-allergic behavior. The SOLPRO server predicted the solubility of vaccine formulations [26].

### 2.4. Prediction and Molecular Docking of Vaccines

Utilizing the Scratch, Galaxy Refine, and Saves services, respectively, the structure of the VC was created, improved upon, and verified [27,28]. In order to dock these constructions into human HLA alleles, the most popular service, Cluspro, was employed (https://cluspro.bu.edu accessed on 20 December 2022) [29].

### 2.5. Simulation of the Docked Complex

Molecular dynamics refer to the general movements of atoms and molecules on a computer. The atoms and molecules are allowed to interact for a defined period of time, revealing information about the dynamic evolution of the system. An analysis was done to determine the stability of the docked complex. MD simulation for the best shortlisted vaccine construct was performed using iMODS [30]. The MD simulation mimics the near-natural environment, such as a cell, for proteins and thus helps to predict the best poses.

### 2.6. C-Immune Simulation

The C-immune tool predictor (https://kraken.iac.rm.cnr.it/C-IMMSIM/ accessed on 25 November 2022) delivers reliable information regarding vaccination strategies [28]. It predicts immunological interactions and makes use of position-specific scoring matrices (PSSM) built from machine learning approaches. This tool was used to assess the immunogenicity and immune response of the modified peptide.

### 2.7. Codon Adaptation and In Silico Cloning

EMBOSS backtranseq (https://www.ebi.ac.uk/Tools/st/emboss_backtranseq/ accessed on 25 November 2022) [31] was used to obtain the DNA sequence from the protein sequence of the constructed vaccine. Subsequently, the JCat server (http://www.jcat.de/ accessed on 25 November 2022) was used for the optimization of the DNA sequence to adapt its codon to most sequenced prokaryotic organisms (*E. coli* K12). Codon optimization analyzes the sequences in light of the cDNA’s GC content, avoiding certain restriction enzyme cleavage sites, bacterial ribosome binding sites, and rho-independent transcription terminators. The expression of the vaccine construct was confirmed with the SnapGene program [32]. The GC content and CAI value were measured for the adapted and un-adapted sequences. The presence of restriction sites in the vaccine construct was investigated in order to clone it into a suitable vector. Finally, the codon-optimized (adapted) DNA sequence of the vaccine was cloned into the *E. coli* pET28a(+) vector using the SnapGene^®^ tool (from Insightful Science; available at https://snapgene.com accessed on 20 December 2022).

## 3. Results

### 3.1. Phylogenetic Analysis of the Retrieved Sequences

The NCBI BlastP search identified 11 protein sequences. All of the 11 proteins were found to be antigenic. The putative virion core protein (lumpy skin disease virus) under accession number QZZ09333.1 was found to have the best antigenicity score of 0.5677 among these 11 proteins (Table 1). These sequences were then aligned using MUSCLE and the phylogenetic tree was constructed (Figure 2).

### 3.2. Physiochemical Properties

The Protparam server identified that the constructed vaccine consists of a total of 349 amino acids, with 41 and 61 as negatively and positively charged residues, respectively [33]. The molecular weight was calculated as 37399.96, whereas the pI was found to be 5.54. The instability index was computed to be 41.40, which classifies the protein as a stable one. The grand average hydropathicity (GRAVY) score was found to be −0.603 and the MT formula was C1611H2687N467O514S1. Lysine was found to be the most prevalent amino acid (15.2%), followed by proline (12.6%), which was followed by arginine (2.3%), glutamate (10.2%), glycine (6.0%), leucine (7.1%), tyrosine and valine (7.2%), serine (6.6%), proline and aspartate (3.9%), threonine (7.4%), tryptophan and methionine (2.4%), and phenylalanine. The VaxiJen v2.0 server predicted the antigenic score of the vaccine construct as 0.5622, which classified the protein as an antigenic protein. The vaccine construct sequence was defined as non-allergenic by the AllerTOP v2.0 server, which further verified its suitability for vaccine construction. SOLpro defined the sequence as soluble with a probability score of 0.918779.

### 3.3. Secondary Structure of Lumpy Skin Protein

PSIPRED was used to predict the secondary structure, showing that our designed vaccine had a high proportion of alpha-helix regions (46.78%) while the lowest percentage corresponded to beta-sheet structures (17.80%). The vaccine exhibited the greatest amino acid content in both the beta sheet (29.88%) and the coil (35.21%) areas, and trRosetta predicted the 3D structure of proteins. (https://yanglab.nankai.edu.cn/trRosetta/ accessed on 20 December 2022) (Figure 3) [34]. Transmembrane topology was predicted using the online program TMHMM. While residues from positions 70–73, 50–55, 101–111, and 220–250 were found to be inside the transmembrane area, residues from positions 111–118, 18–25, 45–58, and 28–31 were revealed to be exposed to the surface. The putative virion core protein was found to have residues from positions 300–205, 309–315, and 330–349 in its core region.

### 3.4. B-Cell Epitope Prediction

B-cell epitopes are either linear or conformational. Proteins have both linear and conformational epitopes. Linear epitopes comprise six to eight contiguous amino acids in the primary amino acid sequence of a polypeptide. Lymphocyte receptors or antibodies recognize linear epitopes in the native, fragmented, or extended conformations of the polypeptide. In contrast, conformational epitopes are created when protein segments are folded into a tertiary structure. The immune system recognizes native conformational epitopes or isolated fragments that retain the appropriate conformational tertiary structure. To find probable B-cell epitopes, the sequence of LSDV was uploaded to the IEDB web service using the default settings. Figure 4 illustrates that the designation “selection of B-cell epitopes” is based on a value greater than the 0.5 threshold. All of the epitopes were checked in the online tools VaxiJen 2.0 and AllerTOP to determine their antigenicity and allergenicity [35,36]. The physicochemical properties are based on the autocross covariance transformation and predicted antigenic epitopes, which were determined independently of alignment. Based on the VaxiJen significance threshold of >0.4, all three outer-membrane proteins were identified as antigenic candidates. Six B-cell epitopes were all anticipated. To determine their antigenicity and allergenicity, all six epitopes were entered into the online calculators VaxiJen 2.0 and AllerTOP. Finally, three epitopes were chosen based on their good antigenic scores and non-allergic behavior; their amino acid sequence, length, and positions are shown in Table 2. The antigenicity investigation revealed that the antigenicity values ranged from 0.5181 to 0.847. According to the results, the lowest value was 0.534, and the highest value was 1.047. However, an average value of 0.62 was observed. 

### 3.5. T-Cell Epitope Prediction

#### 3.5.1. MHC-1 Epitopes 

SMM was used to analyze human IC50 values for MHC and HLA alleles. An IC50 value less than 100 indicates that the epitopes that bind to MHC-1 have a high affinity for binding. An increased affinity for MHC-I molecules is suggested by a lower IC50. The overall number of epitopes was designed to be less than 200, showing enhanced affinity for the alleles. Ten epitopes were selected based on the relationships between MHC-1 alleles and IC50 values. Five distinct epitopes were chosen because they exhibited high antigenic values and both toxicity and non-allergenic properties. The HLA-A*24:02, HLA-A*23:01, HLA-A*02:01, and HLA-A*02:06-specific MHC-1 epitopes were finalized. An antigenic score of 0.9822 applies to CMTKPNKKS (Table 3). 

#### 3.5.2. MHC-II Epitopes 

MHC-II alleles interacted with 550 conserved predicted epitopes with an IC_50_ under 60. Out of 550 epitopes that bind with five alleles, 30 of them were chosen. Finally, based on their allergenicity, toxicity, and antigenicity, four epitopes were chosen for more research. The top binders were determined to be the epitopes LTPQLRTIL, LLGIESVNA, KRGMYKVKT, and ACMTKPNKK, with the alleles HLA-DRB6*01:02, HLA-DRB1*13:03, HLA-DRB1*01:01, and HLA-DRB2*03:01 (Table 4).

### 3.6. Population Coverage

The majority of the world’s population (83.11%) has the MH Class I allele, with the highest percentages found in East Asia (80.07%), Vietnam (61.32%), Pakistan (67.25%), East Africa (72.87%), Poland (71.51%), Mongolia (62.29%), South America (55.41%), the Philippines (70.61%), and India (70.61%); see Figure 5. South America has the lowest value, whereas East Asia has the highest value. East Asia and East Africa have the highest percentage of the MH Class I allele in the population. Different epitopes in MHC-I (YGTMKEGKLE, ACMTKPNKK, and ACMTKPNKKS) are most important for binding. In comparison to the entire world population, two epitope MH Class II alleles (LLGIESVNA and ACMTKPNKK) show significant coverage. The estimated percentage of concentrated population coverage for YGTMKEGKLE was 83.11%. Population coverage data show 87.05% and 77.07% coverage, respectively, for the strong binders to both classes of alleles.

### 3.7. Assessment of the Chimeric Vaccine Construct

A combination of different potent epitopes was made and then tested for its immunogenic potential to design potential multi-epitope vaccine constructs. Adjuvants are short sequences that help increase the immunologic potential of vaccines. These adjuvants are then coupled to the N-terminus of the final prioritized highly antigenic epitopes. Adjuvant sequences will help the designed vaccine generate a stronger immune response Figure 6. The efficacy of the constructs was evaluated with different adjuvants based on their antigenicity and allergenicity. Adjuvants were linked with the aid of EAAAK linkers, and PADRE sequences were employed to overcome polymorphism issues brought on by HLA-DR molecules. PADRE sequences were previously reported to better exhibit the response of cytotoxic T lymphocytes and aid in boosting immune defense [37]. The linkers used to create the construct do not change the conformation of the designed vaccine constructs, as previously reported [38]. The solubility score indicates that the vaccine constructs in *E. coli* cells during heterologous expression are highly soluble. The chimeric vaccine constructs have a solubility score > 0.8 based on SOLpro prediction [39]. The parallel threshold probability was 0.5, which was noteworthy.

### 3.8. Vaccine Refinement and Validation

The 3D structure of the protein can be revealed via protein structure refinement and validation. The ProSA-Web and PROCHECK services are able to confirm protein structural accuracy [40,41]. A total of six models were generated, and the best ones were selected for further evaluation. The Ramachandran plot was calculated using an online tool. Within the Ramachandran plot, all residues are important for protein validation, with varying percentages of most favored (90.1%), allowed (8.0%), generously allowed (1.3%), and disallowed (0.6%) regions. Thus, the designed vaccine has the highest number of most favored regions and was chosen as the final vaccine candidate. The plot confirmed the recommended structure’s high quality and the certainty that the vaccine’s design had areas where immunoglobulins might easily attach. To assess if the provided protein was inside the range of naturally occurring proteins of a comparable size, the Z-score was computed. As illustrated in Figure 7, the input structure’s calculated Z-score of −7.54 is within the typical range for proteins of the same size found in nature.

### 3.9. Molecular Docking of the Vaccine 

Using the Clustpro docking server, protein–protein docking of the final vaccine was performed against TLR-9 (ID: 3WPB) and TLR-4 (ID: 2Z63) in order to find the interaction details. There were 15 models obtained in all. All fifteen docking models were investigated and evaluated using the Pymol application. With respect to the docking of TLR-9-vaccine, the first model was selected among the 15 models, and the TLR-4-vaccine was the second model selected. The TLR-4 vaccine had a binding score of -310.7 and eight H-bond interactions (SER695-ASN481, SER698-ASN481, ARG266-ASP260, GLU135-TYR244, LYS470-SER143, ARG87-TYR244, ARG234-HIS263 and GLN430-LYS146 with distances of 2.86, 2.87, 2.78, 2.83, 2.67, 2.83, and 2, respectively). The docking complexes’ binding site interactions were detailed using the PDBsum online databases. The mechanism of binding interaction of both the complexes, depicted in Figure 8, was assessed using Pymol.

### 3.10. Stability of the Docked Complex

The final docked vaccine–TLR4 complexes were subjected to MD simulation by utilizing the iMODS server. iMODS investigates structures by adjusting various complex force fields and time intervals. In the heat map, interactions between individual residues are represented by lower RMSD values and areas with higher correlation. The docked vaccine and TLR4 complex produced an eigenvalue of 8.60539502 × 10^0^ (Figure 9). Deformability and NMA mobility were represented in Figure 9A,B. Figure 9C,D, which show the variance and B-factor, respectively, and Figure 9E each represented a foreign value and an intricate elastic network.

### 3.11. Immune Simulation 

Molecular dynamics is a powerful technique for analyzing a vaccine’s dependability and stability because it closely relates to the atomic motion of the protein. The C-IMMSIM database determined the vaccine’s immunogenicity, immunological stimulation, and immune response profile. It utilizes machine learning procedures in order to determine immune responses on the basis of compartments, such as bone marrow, thymus, and lymph nodes. The immunogenic profile of the constructed vaccine is depicted in Figure 10. Different IgG and IgM antibodies were detected. Correspondingly, IFN-c and IL-2 were also observed, and effective responses were noted in the HTL and CTL populations (Figure 10).

### 3.12. In Silico Cloning

The CAI value was 0.3296 before codon adaptation and increased to 0.4408 after codon adaptation, indicating that the sequence has a high chance of increased expression. Similarly, the GC content was found to be 68.0216%, which belongs to the optimal range (30–70%) of GC content. Analysis with SnapGene identified two common sites among the expression vector pET28a (+) and the codon-optimized vaccine sequence, including NmeAIII and BspQ1. Both the vaccine sequence and the vector were directionally cloned into their respective cloning sites. After cloning, the final length of the vector and the insert was found to be 3848 bp. The vaccine sequence inserted in the expression vector pET28a (+) is represented in Figure 11.

## 4. Discussion

LSD is a contagious viral illness that frequently manifests as an epizootic in cattle and water buffalo. Because of world trade and an epidemic in the Mediterranean Basin, LSD has recently spread to Asia [42]. Since the start of 2019, when LSD outbreaks were widespread, it has been unclear which strains or variations will be the best match for vaccine production [43]. Until now, capripoxvirus prevention was only achievable with live-attenuated vaccines. Vaccines against Poxviridae viruses, especially smallpox, are known and have been used for many years with a high degree of success. Smallpox vaccines can provide protection against monkeypox and cowpox [44]. The invention of the Poxviridae vaccine has been patented [45]. US7976850B2 (Tapimmune) claims a pharmaceutical composition (vaccine) for the treatment or prophylaxis of Poxviridae infections. The vaccine composition comprises Poxviridae viral antigen, which may be effective to treat many Poxviridae infections, including smallpox, monkeypox, and lumpy virus disease. Our designed vaccine provides good protection against LSD. Based on their research, we believe that our vaccine candidate against monkeypox and smallpox should be studied further. The outcomes of this study highlight the need to explore vaccination failure, particularly vaccine matching and alternative vaccine development. The current research focuses on developing potential vaccine targets against lumpy skin disease, which may be the primary cause of serious sickness in domestic animals, birds, and people [46]. The conventional approach to vaccine development is risky and time-consuming. However, the immunoinformatics-based multi-epitope vaccine approach is an attractive alternative. Surprisingly, the multi-epitope-based vaccine design approach may be a useful and important tactic for preventing pathogenic infection. This approach has the ability to induce cellular immunity due to the presence of various cellular epitopes. Due to these factors, the goal of our study was to develop a potential epitope-based vaccine against the LSD virus utilizing established in silico methods. The NCBI database was used to obtain the FASTA sequence of a putative virion core protein (GenBank: QIN91633.1). The retrieved vaccine was reported to be non-toxic, non-allergenic, and highly antigenic. As a result, we decided to identify epitopes that could be integrated into the vaccine. Then, the prediction of HTL and CTL epitopes was done utilizing several online servers. HTLs are crucial in developing both humoral and cellular immunity [47]. Similarly, CTLs are crucial in the maturation of the adaptive immune response. Epitope-based vaccines have remarkable advantages over conventional ones since they are specific, able to avoid undesirable immune responses, generate long-lasting immunity, and are reasonably cheaper. The B-cell and T-cell epitopes were linked together using different linkers, and our designed vaccines demonstrated a high antigenic score (0.5622) as well as a good solubility score inside *E. coli* (0.581). The physiological properties of the designed vaccine construct were checked, showing the molecular formula (C1611H2687N467O514S18) and indicating that the vaccine had a total of 349 amino acids (349), a molecular weight of 37399.96, and a grand average of hydropathicity of −0.603.

According to secondary structure analysis, our developed vaccine contained the highest proportion of amino acids in the alpha-helix region (46.78%) and the lowest percentage in the beta-sheet structure (17.80%). The beta sheet (29.88%) and coil (35.21%) regions of the vaccine had the most amino acids. The Galaxy Refine and PROCHECK servers were used to refine and validate the vaccines’ three-dimensional structures. The vaccine showed the best validation score, with 90.1% of amino acids located in the most favorable areas of the Rama-plot and just 0.6% in the disallowed regions. The Rama-plot displays the stereochemical statics required for the structure. The results showed that the majority of the residues were contained within the permitted areas. ProSa-Web projected a Z score of −7.54, which was beyond the typical range for naturally occurring proteins of the same size but within the range of 5.26 and 9.5 kcal/mol. Docking was used to examine the binding affinities of the developed vaccine for TLR4 and TLR9. The prioritized global energy values for TLR4 and TLR9 were −310.7 and −236.9, respectively. Additionally, docking by the Cluspro 2.0 server produced more than 25 new structures. The iMODS server was used for molecular dynamic simulation, and the complex vaccine TLR4 was selected because of its energy. High eigenvalues for the vaccine (5.740904 × 10^−7^ for TLR4) indicate a lower likelihood of deformability. High levels of T cytotoxic cells, memory cells, and Ig production were observed, along with an increase in IFN-c and IL-2 levels. This work provided a different vaccination strategy to deal with the antigenic complexity of lumpy skin disease. Based on immunoinformatics approaches, it is thought that the anticipated vaccine is immunogenic and might help eradicate the illness. However, in vitro immunological assays are needed to validate the potency of the vaccine.

## 5. Conclusions

The high burden of lumpy skin disease (LSD), which is associated with the Poxviridae family, is a serious threat to cattle stockbreeding. As such, there is no effective vaccine available for the treatment of LSDV infections. Many antiviral medications have been studied but none have clearly demonstrated effective results against the infection. Reverse vaccinology and computational techniques have been used to build a multi-epitope-based subunit vaccine that could activate humoral and cellular immune responses. This study was undertaken to design a multi-epitope-based vaccine against lumpy skin disease by utilizing several bioinformatics tools. The constructed multi-epitope-based vaccine, coupled with computational analysis techniques such as molecular dynamic simulation, C-immune simulation, codon adaptation, and in silico cloning, validated our design construct as a suitable vaccine candidate. The findings pave the way for the construction of a lumpy skin disease vaccine; however, further experimental validation is needed to confirm the vaccine’s reliability, effectiveness, and safety of the vaccine constructs and to reduce the incidence of lumpy skin disease.

## Figures and Tables

**Figure 1 biomedicines-11-00398-f001:**
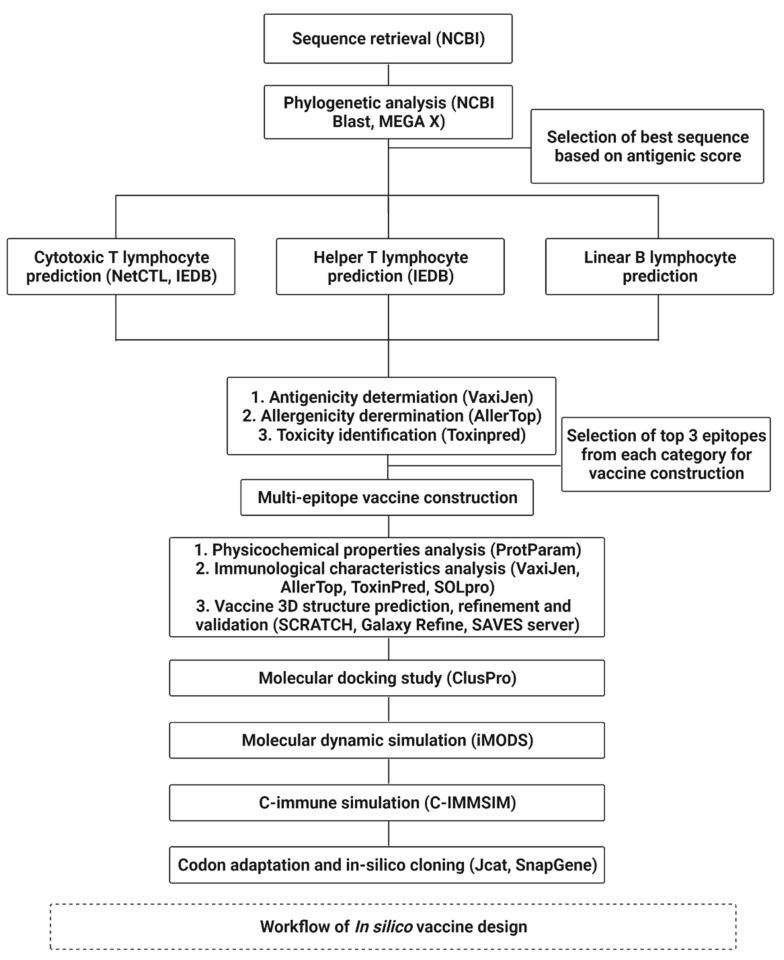
Workflow and tools used in this study for in silico design of a multi-epitope vaccine.

**Figure 2 biomedicines-11-00398-f002:**
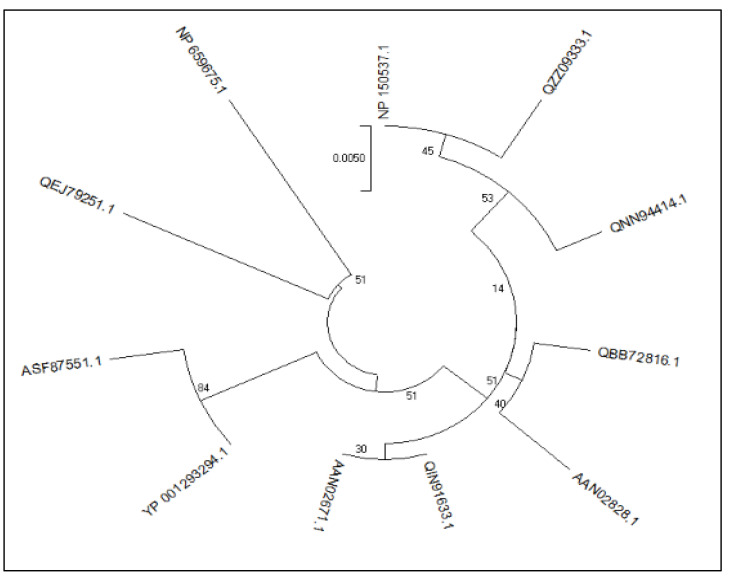
Phylogenetic connections between the investigated protein (QZZ09333.1) and the reference protein.

**Figure 3 biomedicines-11-00398-f003:**
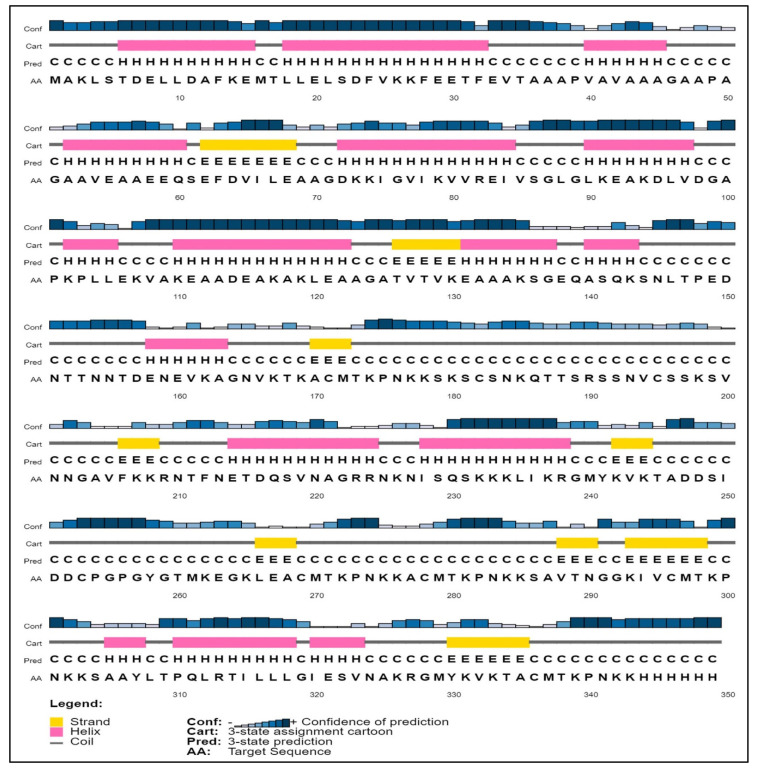
Secondary structure of the target protein predicted graphically.

**Figure 4 biomedicines-11-00398-f004:**
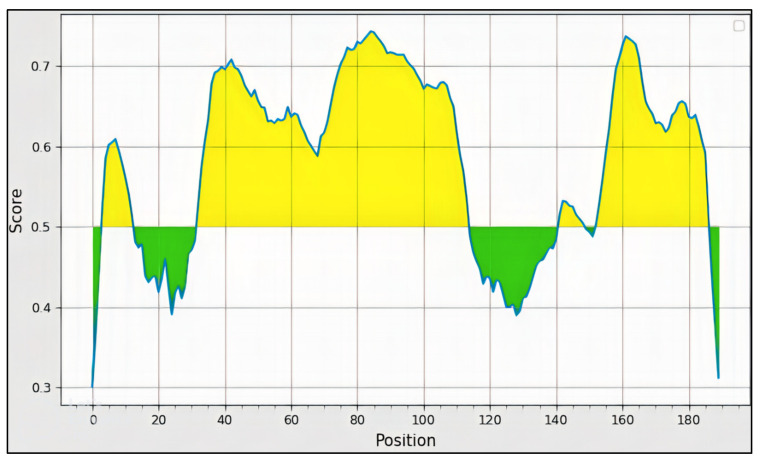
An illustration of a B-cell epitope shows the epitomic region in yellow and the non-epitomic portion in green.

**Figure 5 biomedicines-11-00398-f005:**
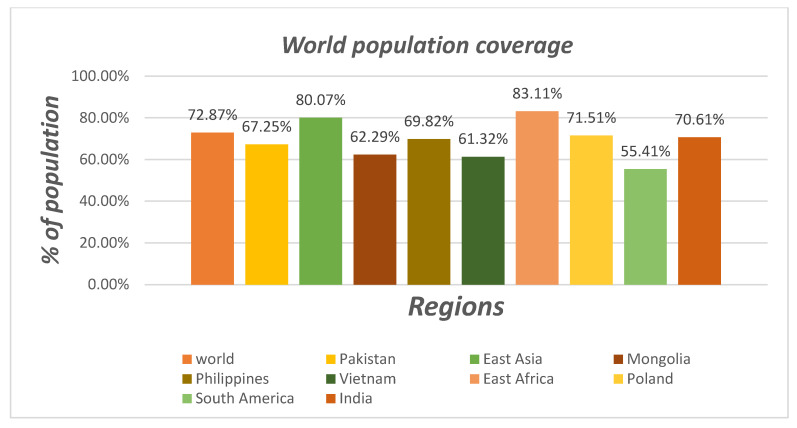
The population coverage of the final 10 epitopes in various parts of the world.

**Figure 6 biomedicines-11-00398-f006:**
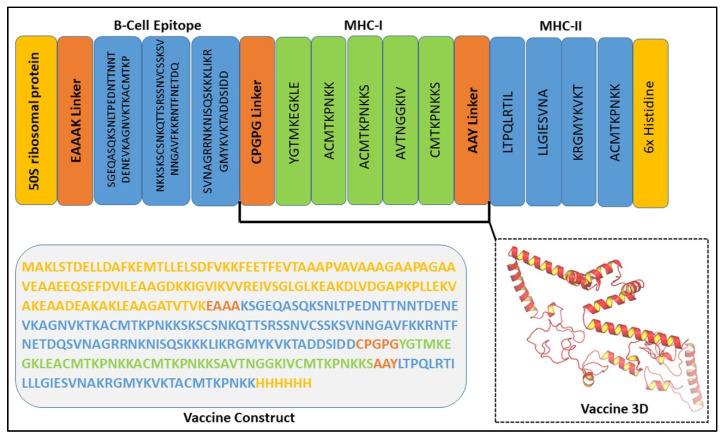
A graphical representation of vaccine in 3D, the adjuvant, and different linkers are shown.

**Figure 7 biomedicines-11-00398-f007:**
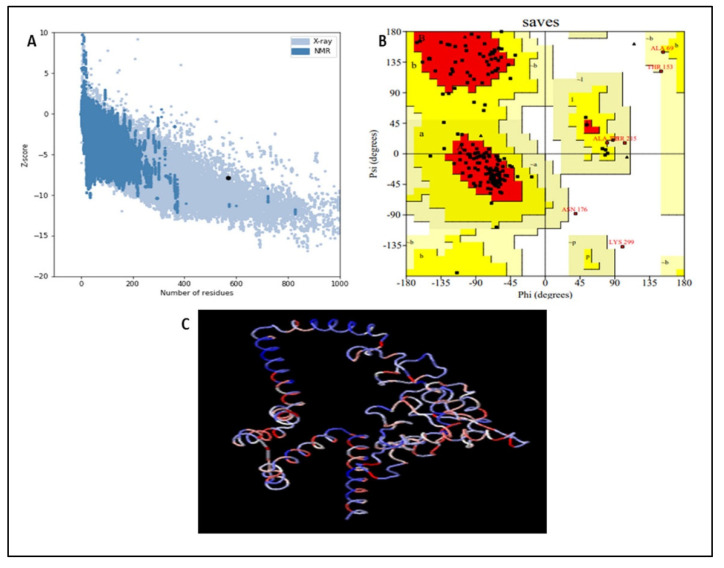
Validation and refinement of the final vaccine using web resources. (**A**): Structure validation with a Z−value of −7.54; R. plot displaying favored (90.1%), allowed (8.0%), generously allowed (1.3%), and disallowed (0.6%) regions. (**B**) Ramachandra plot statistics (**C**) Colors ranging from blue to red indicate increasing residue energy in the 3D representation of vaccine residues.

**Figure 8 biomedicines-11-00398-f008:**
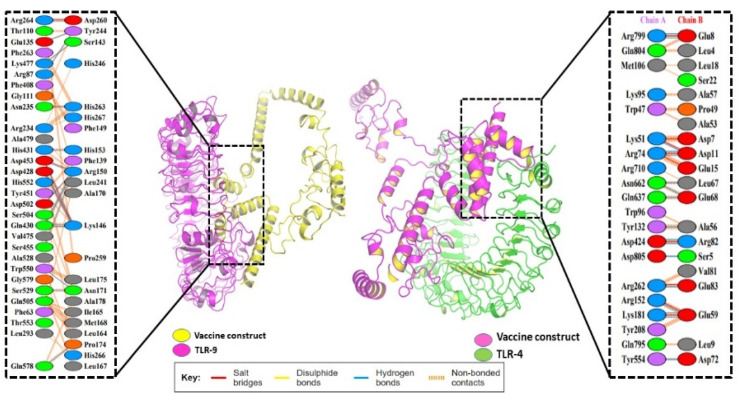
Binding site interaction of both complexes. The PyMOL program was used to assess the binding residues.

**Figure 9 biomedicines-11-00398-f009:**
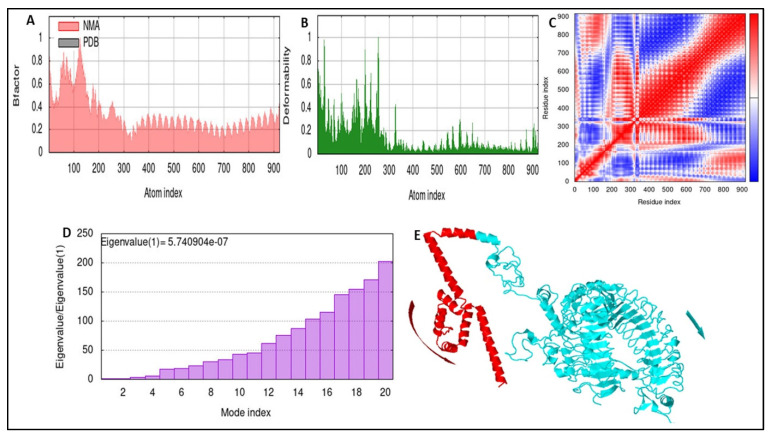
Simulated trajectory of the TLR4 vaccine. (**A**) B-factor and NMA mobilities. (**B**) The ability of molecules to deform each of the residues. (**C**) A covariance matrix displaying both correlated and anticorrelated motion. (**D**) Eigenvalue indicating protein stiffness. (**E**) The vaccine–TLR4 complex after the simulation, showing advanced mobilities.

**Figure 10 biomedicines-11-00398-f010:**
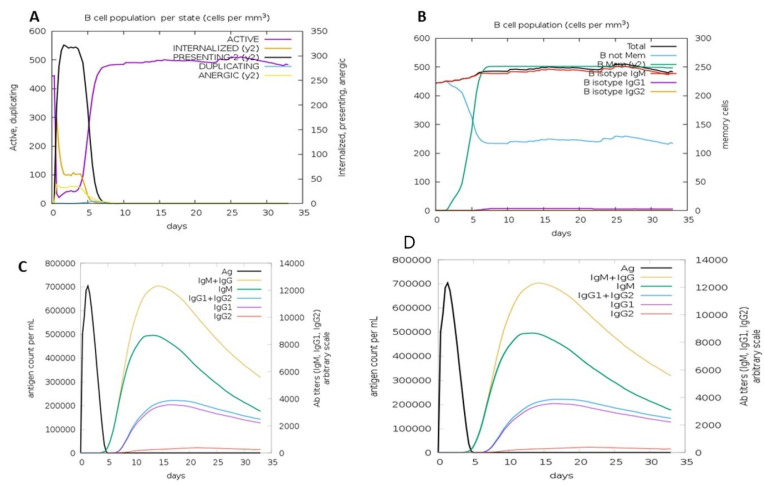
(**A**) Active B cell counts. (**B**) Overall number of B cells, memory cells, and immunoglobulin. (**C**) Antigens and immunoglobulins. Antibodies are subdivided by isotype. (**D**) CD4 T-helper lymphocytes are subdivided by entity-state (i.e., active, resting, anergic, and duplicating).

**Figure 11 biomedicines-11-00398-f011:**
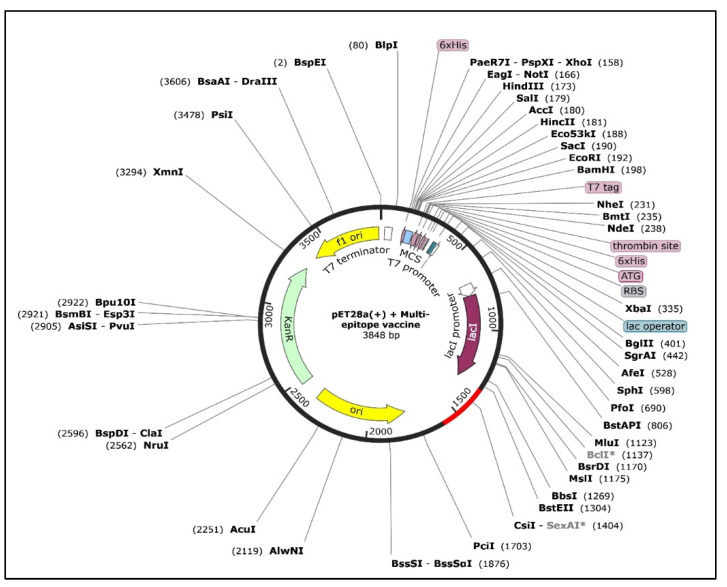
In silico cloning of the vaccine sequence into the pET28a (+) vector, represented in red color (1329 bp to 1589 bp region). The vaccine sequence was inserted between the NmeAIII and BspQ1 sites of the expression vector.

**Table 1 biomedicines-11-00398-t001:** BlastP search yielded protein information for 11 protein sequences from NCBI.

Accession No.	Protein Name	Sequence	VaxiJen Score	Antigenicity
QIN91633.1	putative virion core protein	MSDKKLSRSSYDDYIETINKLTPQLRTILAHISGEQASQKSNLTPEDNTTNNTDENEVKAGNVKTKACMTKPNKKSKSCSNKQTTSRSGNVCSSKSVNNGAVFKKRNTFNETDQIMQAVTNGGKIVYGTMKEGKLEVQGMVGEINQDLLGIESVNAGRRNKNISQSKKKLIKRGMYKVETADDSIDDGMD	0.5504	ANTIGEN
AAN02671.1	putative virion core protein [Lumpy skin disease virus NW-LW]	MSDKKLSRSSYDDYIETINKLTPQLRTILAHISGEQASQKSNLTPEDNTTNNTDENEVKAGNVKTKACMTKPNKKSKSCSNKQTTSRSGNVCSSKSVNNGAVFKKRNTFNETDQIMQAVTNGGKIVYGTMKEGKLEVQGMVGEINQDLLGIESVNAGRRNKNISQSKKKLIKRGMYKVETADDSIDDGMD	0.5504	ANTIGEN
NP_150537.1	LSDV103 putative virion core protein	MSDKKLSRSSYDDYIETINKLTPQLRTILAHISGEQASQKSNLTPEDNTTNNTDENEVKAGNVKTKACMTKPNKKSKSCSNKQTTSRSSNVCSSKSVNNGAVFKKRNTFNETDQIMQAVTNGGKIVYGTMKEGKLEVQGMVGEINQDLLGIESVNAGRRNKNISQSKKKLIKRGMYKVETADDSIDDGMD	0.5529	ANTIGEN
QZZ09333.1	putative virion core protein	MSDKKLSRSSYDDYIETINKLTPQLRTILAHISGEQASQKSNLTPEDNTTNNTDENEVKAGNVKTKACMTKPNKKSKSCSNKQTTSRSSNVCSSKSVNNGAVFKKRNTFNETDQIMQAVTNGGKIVYTMKEGKLEVQGMVGEINQDLLGIESVNAGRRNKNISQSKKKLIKRGMYKVKTADDSIDDGMD	0.5677	ANTIGEN
QBB72816.1	putative virion core protein, partial	MSDKKLSRSSYDDYIETINKLTPQLRTILAHISGEQASQKSNLTPEDNTTNNTDENEVKAGNVKTKACMTKLNKKSKSCSNKQTTSRSGNVCSSKSVNNGAVFKKRNTFNETDQIMQAVTNGGKIVYGTMKEGKLEVQGMVGEINQDLLGIESVNAGRRNKNISQSKKKLIKRGMYKVETADDSIDDGMD	0.5513	ANTIGEN
QNN94414.1	putative virion core protein	MSDKKLSRSSYDDYIETINKLTPQLRTILAHISGEQASQKSNLTPEDNTNNNTDENEVKAGNVKTKACMTKPNKKSKSCS NKQTTSRSSNVCSSKSVNNGAVFKKRNTFNETDQIMQAVTNGGKIVYGTMKEGKLEVQGMVGEINQDLLGIESVNAGRRN KNISQSKKKLIKRGMYKVETADDSIDDGMD	0.5433	ANTIGEN
YP_001293294.1	hypothetical protein GTPV_gp099 [Goat pox virus Pellor]	MSDKKLSRSSYDDYIETINKLTPQLRTILAHISGEQASQKSNLTPEDNTNNNTDENEVKAGNVKTKACITKPNKKSKSCSNKQTTSRSGNVCSSKSVNNGSVFKKRNTFNETDQIMQAVTNGGKIVYGTMKEGKLEVQGMVGEINQDLLGIESVNAGRRNKNISQSKKKLIKRGMYKVETADDSIDDGMD	0.5276	ANTIGEN
AAN02828.1	putative virion core protein	MSDKKLSRSSYDDYIETINKLTPQLRTILAHISGEQASQKSNLTPEDNTNNNTDENEVKAGNVKTKACMTKTNKKSKSCSNKQTTSRSGNVCSSKSVNNGAVFKKRNTFNETDQIMQAVTNGGKIVYGTMKEGKLEVQGMVGEINQDLLGIESVNAGRRNKNISQSKKKLIKRGMYKVETADDSIDDGMD	0.5508	ANTIGEN
ASF87551.1	virion core protein [Goat pox virus]	MSDKKLSRSSYDDYIETINKLTPQLRTILAHISGEQASQKSNLTPEDNTNNNTDENEVRAGNVKTKACITKPNKKSKSCSNKQTTSRSGNVCSSKSVNNGSVFKKRNTFNETDQIMQAVTNGGKIVYGTMKEGKLEVQGMVGEINQDLLGIESVNAGRRNKNISQSKKKLIKRGMYKVETADDSIDDGMD	0.5260	ANTIGEN
QEJ79251.1	core protein [Goat pox virus]	MSDKKLSRSSYDDYIETINKLTPQLRTILAHISGEQTSQKSNLTPEDNTTNNTDENEVKAGNVKTKACITKPNKKSKSCSNKQTTSKSGNVCSSKSVNNGAVFKKRNTFNETDQIMQAVTNGGKIVYGTMKEGKLEVQGMVGEINQDLLGIESVNAGRRNKNISQSKKKLIKRGMYKVETTDDSIDDGMD	0.5648	ANTIGEN
NP_659675.1	Virion core protein [Sheep pox virus]	MSDKKLSRSSYDDYIETINKLTPQLRTILAHISGEQASQKSNLTPEDNTTNNIDENEVKAGNVKTKTCITKPNKKSKSCSNKQTTSRSGNVSSSKSVNNGAVFKKRNTFNETDQIMQAVTNGGKIVYGTMKEGKLEVQGMVGEINQDLLGIESVNAGRRNKNISQSKKKLIKRGMYKVETADDSIDDGMD	0.5394	ANTIGEN

**Table 2 biomedicines-11-00398-t002:** Sequences of selected B-cell epitopes.

Start	End	Peptide	Length
33	72	SGEQASQKSNLTPEDNTTNNTDENEVKAGNVKTKACMTKP	39
80	121	NKKSKSCSNKQTTSRSSNVCSSKSVNNGAVFKKRNTFNETDQ	41
153	187	SVNAGRRNKNISQSKKKLIKRGMYKVKTADDSIDD	35

**Table 3 biomedicines-11-00398-t003:** A list of linear epitopes predicted by the IEDB analysis resource.

Start	End	Peptide	Antigenic Score	Length
4	13	YGTMKEGKLE	0.5095	10
70	78	ACMTKPNKK	0.6486	9
54	63	ACMTKPNKKS	0.5623	10
71	79	AVTNGGKIV	0.4661	9
44	52	CMTKPNKKS	0.9822	9

**Table 4 biomedicines-11-00398-t004:** A list of linear epitopes predicted by the IEDB analysis resource.

Start	End	Peptide	Antigenic Score	Length
18	26	LTPQLRTIL	0.6581	9
64	72	LLGIESVNA	1.4577	9
7	15	KRGMYKVKT	0.4977	9
61	69	ACMTKPNKK	0.6486	9

## Data Availability

Not available.

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
