# Peer review of "An Immunoinformatics Approach to Design Novel and Potent Multi-Epitope-Based Vaccine to Target Lumpy Skin Disease"

_biomedicines, 2023, doi:10.3390/biomedicines11020398_

Round 1

Reviewer 1 Report

Lumpy Skin Disease (LSD) of the Poxviridae family is a serious threat that mostly affects the cattle and causes significant economic loss. LSD has widespread potential and its rapid cross-border spread is also possible.

The study was conducted an epitope-based Lumpy skin vaccine that is efficient, secure, and biocompatible that stimulates both innate and adaptive immune responses using Immunoinformatics techniques.

Thus, the advent of effective vaccines is urgently needed to stop viral transmission through increasing immunological responses. Initially, putative virion core proteins were manipulated, and Cellular epitopes were recognized and connected with the use of adjuvants and linkers.

Numerous bioinformatics methods, including antigenicity testing, transmembrane topology screening, allergenicity assessment, conservancy analysis, and toxicity evaluation, were employed to find superior epitopes. Based on promising vaccine candidates and immunogenic potential, the vaccine design was selected.

Strong interactions between TLR4 and TLR9 and the anticipated vaccine design are revealed by molecular docking.

Finally, Computer simulations were performed based on the high docking score in order to assess the stability and compactness of the constructed vaccine. The simulation outcomes showed the polypeptide vaccine design to be remarkably stable.

The predicted vaccine had High expression, stability, immunogenic qualities, and considerable solubility.

Several computer-based immune response studies confirmed the design vaccine efficacy. Furthermore, according to computer-based analysis, the constructed vaccine provides adequate population coverage and can also be assessed for developing a vaccine against other Poxviridae family virus like the monkeypox virus.

The illustrations of this manuscript are excellent in quality and clear content.

Review. Biomedicines. 2143150

Q 1. Title :

Is short, complete and very descriptive

Q 2. Abstract and Keywords

The Lumpy Skin Disease (LSD) of the Poxviridae family is a serious threat that mostly affects the cattle and causes significant economic loss. LSD has widespread potential and its rapid cross-border spread is also possible. The study was conducted an epitope-based Lumpy skin vaccine that is efficient, secure, and biocompatible that stimulates both innate and adaptive immune responses using Immunoinformatics techniques. Thus, the advent of effective vaccines is urgently needed to stop viral transmission through increasing immunological responses. Initially, putative virion core proteins were manipulated, and Cellular epitopes were recognized and connected with the use of adjuvants and linkers. Numerous bioinformatics methods, including antigenicity testing, transmembrane topology screening, allergenicity assessment, conservancy analysis, and toxicity evaluation, were employed to find superior epitopes. Based on promising vaccine candidates and immunogenic potential, the vaccine design was selected.

Strong interactions between TLR4 and TLR9 and the anticipated vaccine design are revealed by molecular docking. Finally, Computer simulations were performed based on the high docking score in order to assess the stability and compactness of the constructed vaccine. The simulation outcomes showed the polypeptide vaccine design to be remarkably stable. The predicted vaccine had High expression, stability, immunogenic qualities, and considerable solubility. Several computer-based immune response studies confirmed the design vaccine efficacy. Furthermore, according to computer-based analysis, the constructed vaccine provides adequate population coverage and can also be assessed for developing a vaccine against other Poxviridae family virus like the monkeypox virus.

Keywords are well selected, appropriated and in enough number

Q 3. 1. Introduction

The Lumpy Skin Disease (LSD), is associated with Poxviridae family, is a serious threat to cattle stockbreeding. Around the globe mainly Asia, Europe, and the Middle East, the infection has become a deadly concern for large domesticated ruminants. According to the world organization for animal health (WOAH), Because of the massive financial failures and the possibilities for spread, it has been identified and added to the notifiable disease list, because of its possibility for rapid cross-border spread.

Animals of all ages and breeds are impacted, although the young and those lactating at their peak are most vulnerable. Because it causes a sharp decrease in milk production, abortions, poor coat health, and sterility in bulls, the disease is devastating. It spread so quickly across continents if infected animals are moved between farms and quarantine protocols are relaxed . However, there is no epidemiological proof that the disease is contagious among animals . The disease was confined to larger Africa until 1988 before gradually spreading to the Middle East, Eastern Europe, and ultimately the Russian Federation. New cases were then recorded in South and East Asia in 2019 as the pandemic continued to expand. The reappearance of the lumpy skin virus in different regions of the world highlighted the necessity of reconsidering the nature of the illness, the viral transmission mechanism, and modernized preventative and adaptive control approaches.

A comprehensive assessment of LSD has been done, primarily focusing on the South-Eastern region of Asia, taking into account the aforementioned facts. LSD is a double-stranded DNA virus that is roughly 150 kb in size, a member of the Capripoxvirus genus, and has genetic ties to the sheep pox and goat pox viruses. The virus maintains a constant pH between 6.6 and 8.6 and is typically resistant to numerous physical and chemical agents, but it is inclined to more alkaline environments. It only has the potential to survive for thirty three days in necrotic skin nodules, thirty five days in desiccated crusts, 6 months in sunlight-protected diseased tissue, and at least 18 days in air-dried hides at room temperatura.

The standard method of developing vaccines is costly and time-consuming. The multi-epitope vaccine method based on reverse vaccinology an attractive option, nevertheless. To combat human diseases caused by Monkey Pox, Burkholderia pseudomallei , SARS-CoV-2, Mokola Rabies virus , and Congo virus, various vaccines are being created using reverse vaccinology techniques . Herein, we developed an epitope-based Lumpy skin vaccine that is efficient, secure, and biocompatible that stimulates both specific and Non-specific immune responses. In this case, we used immunoinformatics to create the most antigenic and non-allergic vaccine epitopes. The discoveries pave the foundation for the development of a lumpy skin vaccine, but additional experimental verification is required to lower the lumpy skin disease burden.

Methodology

The overall mechanism and various tools used in this study for designing a multiepitope vaccine by in silico processes are depicted in figure 1.

Sequence data retrieval and Phylogenetic

Sequences were obtained from publicly available database NCBI using the accession number (QIN91633.1). The non-redundant sequences were obtained via NCBI blast tools and were subjected to alignment through MUSCLE program and the top 11 sequences (along with ref seq) were collected using FASTA file. Then the non-redundant Fasta file was used for phylogenetic tree analysis and was constructed using Mega-X. All of the obtained sequences were assessed for antigenicity and allergenicity using VaxiJen v2.0

(http://www.ddg-pharmfac.net/vaxijen/VaxiJen/VaxiJen.html) and AllerTOP v2.0 (https://www.ddg-pharmfac.net/AllerTOP/) server, respectively. The threshold parameter of the VaxiJen server was set to 0.4, throughout this study. The protein with highest antigenicity and non-allergen property was selected for further analysis.

B and T cell lymphocytes prediction

High-antigenicity and non-allergic sequences are subjected to antibody recognition, and such sequences may be used to predict B-cell epitopes . The smallest immunogenic peptides found in an antigen, known as an epitope, have the capacity to activate the immune responses . The epitopes for immunological responses were predicted using the IEDB Resource (http://tool.iedb.org/main/) . T-cell epitope binding scores with MHC-I and MHC-II were computed. The final selection was chosen when the IC50 values were < 100. The lesser the IC50 the more will be the binding affinity.

Development vaccine based on cytotoxicity, allergenicity, and antigenicity.

The VaxiJen server performed the antigenicity assessment with an accuracy percentage of between 70 and 89 %. The antigenic calling threshold was set at > 0.5 VaxiJen probability score, as previously stated. The allergenicity and toxicity of the epitopes were predicted using AllerTOP and ToxinPred, respectively. The construction of the multi-epitope vaccine was done using the mentioned standards. To create multi-epitope vaccination constructs, the lead epitopes were linked together using adjuvants, and amino acid linkers (such as 50S ribosomal proteins and CPGPG and AAY) . The HLA DRbinding epitope was also with the adjuvant in order to improve the efficacy . Low toxicity or allergenicity and high immunogenicity were examined in the constructions made up of various combinations. AllerTOP anticipated the non-allergic behavior. The SOLPRO server predicts the solubility of vaccine constructions

Prediction and molecular docking of vaccines

Utilizing the Scratch, Galaxy Refine, and Saves services, respectively, the structure of the VC was created, improved upon, and verified . In order to dock these constructions into human HLA alleles, the most popular Cluspro service was employed (https://cluspro.bu.edu) .

Simulation of the docked complex

Molecular dynamics refers to the general movements of atoms and molecules on a computer. The atoms and molecules are allowed to interact for a defined period of time, revealing information on the dynamic evolution of the system. Analysis was done to determine the stability of the docked complex. MD simulation for the best shortlisted vaccine construct was performed using iMODS . The MD simulation mimics the near natural environment like a cell for proteins and thus helps to predict the best poses.

C-immune simulation

T C-immune tool predictor (https://kraken.iac.rm.cnr.it/C-IMMSIM/) to delivers reliable information regarding vaccination strategy . It predicts immunological interactions that makes use of position-specific scoring matrices (PSSM) built from machine learning approaches. To assess the immunogenicity and immune response of the modified peptide.

Codon adaptation and in-silico cloning

EMBOSS backtranseq (https://www.ebi.ac.uk/Tools/st/emboss_backtranseq/) was used to obtain the DNA sequence from the protein sequence of the constructed vaccine. Subsequently, the JCat server (http://www.jcat.de/) was used for the optimization of the DNA sequence to adapt its codon to most sequenced prokaryotic organisms (E. coli K12). Codon optimization analyzes the sequences in light of the cDNA's GC content. Avoiding certain restriction enzyme cleavage sites, bacterial ribosome binding sites, and rho-independent transcription terminators. Employing the Snapgene program, the expression of the vaccine construct was confirmed. GC content and CAI value were measured for the adapted and un-adapted sequences. The presence of restriction sites in the vaccine construct was investigated in order to clone it to a suitable vector. Finally, the codon-optimized (adapted) DNA sequence of the vaccine was cloned into the E. coli pET28a(+) vector using the SnapGene® tool (from Insightful Science; available at https://snapgene.com).

Results

Phylogenetic analysis of the retrieved sequences

The NCBI BlastP search identified 11 protein sequences. All of the 11 proteins were found antigenic. The putative virion core protein [Lumpy Skin Disease Virus] under the accession number QZZ09333.1 was found to have the best antigenicity score of 0.5677 among all these 11 proteins (Table 1). These sequences were then aligned using MUSCLE, and the phylogenetic tree was constructed (Figure 2).

Physiochemical Properties

The Protparam server identified that the constructed vaccine consists of a total number of 349 amino acids, with 41 and 61 as negatively charged residues and positively charged residues, respectively. The molecular weight was calculated as 37399.96 whereas the pI was found to be 5.54. The instability index was computed to be 41.40, which classifies the protein as a stable one. The grand average of hydropathicity (GRAVY) score was found to be -0.603 and the MT formula C1611H2687N467O514S1. Among the amino acid composition, Lysine was found to be more prominent (15.2%), proline (12.6%), which was followed by Arginine (2.3%), glutamate (10.2%), glycine (6.0%), leucine (7.1%), tyrosine and valine (7.2%), serine (6.6%), proline and aspartate (3.9%), threonine (7.4%), tryptophan and methionine (2.4%), phenylalanine and asparagine (2.0%), isoleucine (1.6%), glutamine (0.8%), arginine and histidine (0.4%). The VaxiJen v2.0 server predicted the antigenic score of the vaccine construct as 0.5622 which classified the protein as an antigenic protein. The vaccine construct sequence was defined as non-allergen by AllerTOP v2.0 server further verified its suitability for vaccine construction. SOLpro defined the sequence as soluble with a probability score of 0.918779.

Secondary Structure of Lumpy Skin Protein

PSIPRED was used to predict secondary structure, showing that our designed vaccine had the high proportion of alpha-helix region (46.78%) while having the lowest percentage in the beta-sheet structure (17.80%). The vaccine exhibited the greatest amino acid content in both the beta sheet (29.88%) and the coil (35.21%) areas, and trRosetta predicted the 3D structure of proteins  (https://yanglab.nankai.edu.cn/trRosetta/) (Figure 3)

Transmembrane topology was predicted using the online program TMHMM. While residues from positions 70–73, 50–55, 101–111, and 220-250 were discovered inside the transmembrane area, residues from positions 111–118, 18–25, 45–58, and 28–31 were revealed to be exposed to the surface. The putative virion core protein was found to have residues from positions 300–205, 309–315, and 330–349 in its core region.

B-cell epitope prediction

To find probable B-cell epitopes, the sequence of LSD was uploaded to the IEDB web service using the default settings. Figure 4 illustrates the designation of Selection of B-cell epitopes is based on a value greater than the 0.5 threshold. Six B-cell epitopes were all anticipated. To determine their antigenicity and allergenicity, all six epitopes were entered into the online calculators VaxiJen 2.0 and AllerTOP. Three of the six epitopes were chosen as the final three based on good antigenic scores and non-allergic behavior; their amino acid sequence, length, and positions are shown in Table 1. The antigenicity investigation revealed that the antigenicity value ranged from 0.5181 to 0.847. However, an average value of 0.62 was observed.

Epitopes predicted from vaccine candidates

B-cell epitopes were predicted using a 0.5 threshold from FASTA sequences of putative virion core protein proteins. All of the epitopes were entered into the online tools VaxiJen 2.0 and AllerTOP to determine their antigenicity and allergenicity. The physicochemical behavior of this server is dependent on autocross covariance transformation and alignment-independent predicted antigenic epitopes. Based on the VaxiJen significant threshold of > 0.4, all three outer-membrane proteins were identified as antigenic candidates. Three epitopes were chosen from among the five based on good score, and their a.a sequence, length, and position are listed in Table 1. According to the result, the lowest value was recorded 0.534, and the highest value was 1.047. Furthermore, a mean of 0.72 was recorded.

T-Cell Epitope Prediction

MHC-1 Epitopes

SMM was used to analyze human IC50 values for MHC and HLA alleles. An IC50 value less than 100 indicates that the epitopes that bind with MHC-1 have a strong and greater binding affinity. An increased affinity for MHC-I molecules is suggested by a lower IC50. The overall number of epitopes was designed to be less than 200 showing enhance affinity toward the alleles. Ten epitopes were selected based on the relationships between MHC-1 alleles and IC50 values. Five distinct epitopes were chosen because they exhibited Non-allergic, high antigenic value as well as toxicity. The HLA-A*24:02, HLAA*23:01, HLA-A*02:01, and HLA-A*02:06-specific MHC-1 epitopes have been finalized. An antigenic score of 0.9822 applies to CMTKPNKKS (Table 2).

MHC-II Epitopes

MHC-II alleles interacted with 550 conserved predicted epitopes with IC50 under 60. Out of 550 epitopes that bind with five alleles, 30 of them chosen. Finally, based on their allergenicity, toxicity, and antigenicity, 4 epitopes were chosen for more research. The top binder was determined to be the epitopes LTPQLRTIL, LLGIESVNA, KRGMYKVKT, and ACMTKPNKK, with the alleles HLA-DRB6*01:02, HLA-DRB1*13:03, HLA-DRB1*01:01, and HLA-DRB2*03:01 (Table 4).

Population Coverage

The majority of the world's population (83.11%) has the MH Class-I allele, with the highest percentages found in East Asia (80.07%), Vietnam (61.32%), Pakistan (67.25%), East Africa (72.87%), Poland (71.51%), Mongolia (62.29%), South America (55.41%), the Philippines (70.61%), and India (70.61%), Figure 5. In South America, shows the lowest value whereas east Asia shows high value. East Asia and East Africa have the highest percentage alleles in the population, along with East Asia. Defferent epitopes in MHC-I (YGTMKEGKLE, ACMTKPNKK and ACMTKPNKKS), are most important for binding . In comparison to the entire world population, two epitope MH Class-II alleles (LLGIESVNA and ACMTKPNKK) show significant coverage. The estimated percentage of concentrated population coverage for YGTMKEGKLE was 83.11%. Population coverage data show 87.05% and 77.07% coverage, respectively, for the strong binders to both class of alleles.

Assessment of the chimeric vaccine construct

The combination of different potent epitopes was made and then tested for their immunogenicity potential to design potential multi-epitope vaccine constructs. Adjuvants are short sequences that help to increase the immunologic potential of vaccine. Here we coupled these adjuvants to the N-terminal of the final prioritized highly antigenic epitopes. Adjuvant sequences will help the designed vaccine to generate stronger immune response. The efficacy of the constructs was evaluated with different adjuvants based on their antigenicity and allergenicity. Adjuvants were linked with the aid of EAAAK linkers, and PADRE sequences were employed to get over polymorphism issues brought on by HLA-DR molecules. PADRE sequences were reported previously to better exhibit response of Cytotoxic T lymphocytes aid in boosting immune defense [36]. The linkers used to create construct do not change their conformation of designed vaccine constructs, as reported [37]. The solubility score infers highly soluble property of vaccine constructs in E. coli cell during heterologous expression. The chimeric vaccine constructs have solubility score > 0.8 based on

Vaccine refinement and validation

The 3D structure of the protein can be revealed via protein structure refinement and validation. The ProSA-Web and PROCHECK services that confirm the protein structural accuracy [39, 40]. The total of six models were generated, and the best models were selected for further evaluation. The Ramachandran plot was calculated using the online tool. Within the Ramachandran plot, all residues are most important for protein validation, with the most favored (90.1%), allowed (8.0%), generously allowed (1.3%), and disallowed (0.6%) regions, respectively. Thus, the designed vaccine has the highest number of most favored regions and was chosen as a final vaccine candidate. It confirmed the recommended structure's high quality and the certainty that the vaccine's design had areas where immunoglobulins might easily attach. To assess if the provided was inside of the range of naturally occurring proteins of a comparable size, the Zscore was computed. As illustrated in Figure 7, the input structure's calculated Z-score of -7.54, the typical range for proteins of the same size found in nature.

Molecular docking of vaccine

Using the Clustpro docking server, protein-protein docking of the final vaccine was performed against TLR-9 (ID; 3WPB) and TLR-4 (ID; 2Z63) in order to find the interaction details. There were 15 models obtained in all. All fifteen docking models were investigated and evaluated using the Pymol application. Docking of TLR-9-vaccine: the first model was selected among the 15 models, and TLR-4-vaccine was the second model selected, The TLR-4 vaccine displayed a docking result with a binding score of -310.7 and a total of eight H-bond interactions (SER695-ASN481, SER698-ASN481, ARG266-ASP260, GLU135-TYR244, LYS470-SER143, ARG87-TYR244,ARG234-HIS263 and GLN430-LYS146 with distances of 2.86Å , 2.87Å , 2.78Å , 2.83Å , 2.67Å , 2.83Å , and 2.78Å ,) while the TLR-9 vaccine had a binding score of -236.9 and a total of four H-bond interactions (ASP805SER5, ARG799-GLU8, LYS51-ASP11, ARG74-ASP11 with distances 2.92 Å ,3.13 Å ,2.68 Å and 2.53 Å ). The PDBsum online databases were used to provide a detailed binding site interaction of the docking complexes. The mechanism of binding interaction of both the complexes depicted in Figure 8 was assessed using Pymol.

Stability of the docked complex

The final docked vaccine-TLR4 complexes were subjected to MD simulation by utilizing the iMODS server. iMODS investigate structures by adjusting various complex force fields and time intervals. In the heat map, lower RMSD and higher co-related areas represented interactions between individual residues. The docked complex of vaccine and TLR4 produced an eigenvalue of 5.740904e-07 (Figure 9). Deformability and NMA mobility were represented in Figure 9 (A and B). Figures C and D, which show the variance and B-factor, respectively, and Figures E and F, respectively, each represented a foreign value and an intricate elastic network.

Immune Simulation

Molecular dynamics is a powerful technique for analyzing the vaccine's dependability and stability because it closely relates the atomistic motion of the protein. The CIMMSIM database determined the vaccine's immunogenicity, immunological stimulation, and immune response profile identification. It utilizes machine-learning procedures in order to determine immune responses on the basis of compartments, such as bone marrow, thymus, and lymph nodes. The immunogenic profile of the constructed vaccine is depicted in Figure 10. Immune simulation results showed that after primary responses to antigens IgG1 + IgG2, IgM + IgG, IgG2, and IgM having effective antibody titers is depicted in Figure 10A, and effective responses were noted in the HTL and CTL populations (Figure 10B).

In silico cloning

The CAI value was 0.3296 before the codon adaptation and increased to 0.4408 after the codon adaptation, indicating that the sequence has a high chance of increased expression. Similarly, the GC content has been found to be 68.0216%, which belongs to the optimal range (30-70%) of GC content. Analysis with Snapgene has identified two common sites between the expression vector pET28a (+) and the codon-optimized vaccine sequence, including NmeAIII and BspQ1. Both the vaccine sequence and the vector were directionally cloned into their respective cloning sites. After cloning, the final length of the vector and the insert was found to be 3848 bp. The vaccine sequence inserted in the expression vector pET28a (+) is represented in Figure 11.

Discussion

LSD is a contagious viral illness that frequently manifests as an epizootic in cattle and water buffalo. Because of world trade and an epidemic in the Mediterranean Basin, LSD has recently spread to Asia. Since the start of 2019, when LSD outbreaks were widespread, it has been unclear which strains or variations will be the best match for vaccine production. Until now, Capripoxvirus prevention was only achievable with live-attenuated vaccines. Vaccines against Poxviridae viruses, especially smallpox, are known and have been used for many years with a large degree of success. The smallpox vaccines can provide protection against monkeypox and cowpox. The invention of the Poxviridae vaccine has been patented. US7976850B2 (Tapimmune) claims a pharmaceutical composition (vaccine) for the treatment or prophylaxis of Poxviridae infections.

The vaccine composition comprises Poxviridae viral antigen, which may be effective to treat many Poxviridae infections, including smallpox, monkeypox as well as a lumpy virus disease. Our designed vaccine provides good protection against LSD. Based on literature teaching, the authors also opine to further investigate our vaccine candidate against monkeypox and smallpox. The outcomes of the study highlight the need to explore vaccination failure, particularly vaccine matching and alternative vaccine development. The current research focuses on developing potential vaccine targets against Lumpy Skin Disease, which may be the primary cause of serious sickness in domestic animals, birds, and people. Surprisingly, the multi-epitope-based vaccine-design approach may be a useful and important tactic for preventing pathogenic infection. Due to the presence of different cellular epitopes, have the affinity to induces cellular immunity.

Due to these factors, the goal of our study was to develop a potential epitope-based vaccination against the LSD virus utilizing established in silico methods. The NCBI database was used to obtain the FASTA sequence of a putative virion core protein (GenBank: QIN91633.1). The retrieved vaccine was reported to be non-toxic, non-allergenic, and highly antigenic. As a result, it was decided to identify epitopes that could be integrated into the vaccine.

Then, the prediction of HTL and CTL epitopes was done utilizing several online servers. HTLs are crucial in developing both humoral and cellular immunity. Similarly, CTLs are crucial in the maturation of the adaptive immune response. The B-cell and T-cell epitopes were linked together using different linkers, and the efficacy of our designed vaccines demonstrates a high antigenic score (0.5622) as well as a good solubility score inside E. coli (0.581). The physiological properties of the designed vaccine construct were checked, showing a molecular formula (C1611H2687N467O514S18) containing the total number of amino acids (349), the molecular weight (37399.96), and the grand average of hydropathcity (-0.603).

According to secondary structure analysis, our developed vaccine contained the highest proportion of amino acids in the alpha-helix region (46.78%) and the lowest percentage in the beta-sheet structure (17.80%). The beta sheet (29.88%) and coil (35.21%) regions of the vaccine had the most amino acids. The Galaxy Refine and PROCHECK servers were used to refine and validate the vaccine constructions' three-dimensional structures. Vaccine showed the best validation score, with 90.1% of amino acids located in the most favorable areas of the Rama-plot and just 0.6% in the disallowed regions. The Ramaplot displays the requisite stereochemical statics required for the structure. Results showed that the majority of the residues were contained within the permitted areas. ProSa-Web projected a Z score of -7.54, which was beyond the typical range for naturally occurring proteins of the same size but within the range of 5.26 and 9.5 kcal/mol. Docking was used to examine the binding affinities of the developed vaccine and TLR4 and TLR9. Prioritized global energy values for TLR4 and TLR9 were -310.7 and -236.9, respectively.

Additionally, docking by the Cluspro 2.0 server produced more than 25 new structures. The iMODS server was used for molecular dynamic simulation, the complex vaccine TLR4 was selected because of its energy. High eigenvalues for the vaccine (TLR4 and 5.740904e07, respectively) indicate a lower likelihood of deformability. The immune simulation graph shows that our designed vaccine has a significant level of IgM generation after inoculation, suggesting the primary response. A rise in immunoglobulin expression in the B-cell also contributed to a decrease in the antigen concentration. This work provided a different vaccination strategy to deal with antigenic complexity of lumpy skin disease. Based on immunoinformatics approaches, it is thought that the anticipated vaccine is immunogenic and might help eradicate the illness. To confirm the vaccine's efficacy, in vitro immunological studies are necessary.

References are well selected, recently published and in a good number

Author Response

Dear reviewer! Please see the attachment

Reviewer #1

 The illustrations of this manuscript are excellent in quality and clear content
Q 1. Title:

Is short, complete and very descriptive

Q 2. Abstract and Keywords

Keywords: are well selected, appropriated and in enough number

References; are well selected, recently published and in a good number

Response:

Thank you very much and we appreciate the time and efforts spent on our former manuscript and we also appreciate the comments.

Reviewer 2 Report

The authors have done a detailed in-silico analysis. However, this is not enough to predict the efficacy of a vaccine. Authors must carry out some experimental work in the wet lab, by adding in vitro and in vivo data.

I am afraid the manuscript can not be accepted in its current form.

Author Response

Reviewer #2

The authors have done a detailed in-silico analysis. However, this is not enough to predict the efficacy of a vaccine. Authors must carry out some experimental work in the wet lab, by adding in vitro and in vivo data.

I am afraid the manuscript cannot be accepted in its current form.

Response: To prevent uncontrollable spread, more intensive surveillance, and research on lumpy skin disease biology, history, transmission mode, host interactions, drug targets and vaccine development are extremely crucial. Governments, academics, industries and healthcare systems work together to achieve this goal. The conventional approach to vaccine development is risky and time-consuming. However, the immunoinformatics based multi-epitope vaccine approach is an attractive alternative.

Reviewer 3 Report

In this predominantly theoretical manuscript, an extensive and multi-faceted immunoinformatics approach is used to design a multi-epitope vaccine for Lumpy skin disease, a poxviruses that mostly affects cattle and has become an important economic burden.  The study utilizes several bioinformatics methods, including antigenicity testing, transmembrane topology screening, allergenicity assessment and toxicity testing to identify highly effective epitopes.  The authors are to be commended for the completeness of the study.  They have clearly used everything at their disposal to maximize the vaccine’s chances for success.  There are no problems identified with respect to solubility, immunogenicity or stability.  As judged by everything presented here, the final product has been sufficiently vetted is just about every way possible to ensure that it will be successful.

The manuscript is considered technically and scientifically solid.  However, the presentation itself is lacking in several regards: 

First, the usage of the English language and grammar is quite weak and requires considerable attention. 

Second, sections of the text are extremely confusing, although in some cases it may be merely be a case of referencing the incorrect figure or table.  At the very bottom of page 8, Table 1 is said to show the B cell epitopes, but it does not.  The first paragraph under the “Epitopes predicted from vaccine candidates” section on page 9 is very confusing.  It is said that three epitopes were chosen among the five.  What five?  I got lost here.  In the last sentence on page 10, the antigenic score for an epitope is given and Table 2 in referenced.  But, I believe it is in Table 3.  There are numerous other examples where more attention to detail is required for clarity.  These instances detract from the importance of the study. 

Third, in Table 2, residues 14-55 are listed as the site of a B cell epitope.  However, in Figure 4, this region is predicted to be non-epitomic.

Four, near the end of the Discussion, the statement is made that the designed vaccine generates a significant IgM response. But, IgM antibodies are known to be low affinity binders.  Isn’t it a problem that IgG antibodies are not elicited, as these are likely to be the most effective?

Fifth, this vaccine by definition encodes linear B cell epitopes.  Whereas most B cell epitopes are conformational in structure, the fact that the vaccine encodes only linear ones is considered a relative weakness.  While realizing this cannot be avoided, the authors should still acknowledge this at some point in the manuscript.

Author Response

Reviewer #3

In this predominantly theoretical manuscript, an extensive and multi-faceted immunoinformatics approach is used to design a multi-epitope vaccine for Lumpy skin disease, a poxvirus that mostly affects cattle and has become an important economic burden.  The study utilizes several bioinformatics methods, including antigenicity testing, transmembrane topology screening, allergenicity assessment and toxicity testing to identify highly effective epitopes.  The authors are to be commended for the completeness of the study.  They have clearly used everything at their disposal to maximize the vaccine’s chances for success.  There are no problems identified with respect to solubility, immunogenicity or stability.  As judged by everything presented here, the final product has been sufficiently vetted is just about every way possible to ensure that it will be successful.

The manuscript is considered technically and scientifically solid.  However, the presentation itself is lacking in several regards: 

Comments: First, the usage of the English language and grammar is quite weak and requires considerable attention. 

Response:

According to the reviewer comment we improved the English language and grammar portion.

Comments: Second, sections of the text are extremely confusing, although in some cases it may be merely be a case of referencing the incorrect figure or table.  At the very bottom of page 8, Table 1 is said to show the B cell epitopes, but it does not.  The first paragraph under the “Epitopes predicted from vaccine candidates” section on page 9 is very confusing.  It is said that three epitopes were chosen among the five.  What five?  I got lost here.  In the last sentence on page 10, the antigenic score for an epitope is given and Table 2 in referenced.  But, I believe it is in Table 3.  There are numerous other examples where more attention to detail is required for clarity.  These instances detract from the importance of the study. 

Response:

Thank you so much for your valuable comments! I corrected and updated the manuscript according to the highlighted comment.

Comments: Third, in Table 2, residues 14-55 are listed as the site of a B cell epitope.  However, in Figure 4, this region is predicted to be non-epitomic.

Response:

I corrected it according to the reviewer comment.

Comments: Four, near the end of the Discussion, the statement is made that the designed vaccine generates a significant IgM response. But, IgM antibodies are known to be low affinity binders.  Isn’t it a problem that IgG antibodies are not elicited, as these are likely to be the most effective?

Response:

I corrected it according to the reviewer comment.

Comments: Fifth, this vaccine by definition encodes linear B cell epitopes.  Whereas most B cell epitopes are conformational in structure, the fact that the vaccine encodes only linear ones is considered a relative weakness.  While realizing this cannot be avoided, the authors should still acknowledge this at some point in the manuscript.

Response:

Epitopes are either linear or conformational. Proteins have both linear and conformational epitopes. Linear epitopes comprise six to eight contiguous amino acids in the primary amino acid sequence of a polypeptide. Lymphocyte receptors or antibodies recognize linear epitopes in the native, fragmented, or extended conformations of the polypeptide. In contrast, conformational epitopes are created when protein segments are folded into a tertiary structure. The immune system recognizes the native conformational epitopes or the isolated fragments that retain the appropriate conformational tertiary structure.

Reviewer 4 Report

The manuscript by Shahab M et al. entitled “An Immunoinformatics approach to design Novel and potent Multi-Epitope based vaccine to target Lumpy skin disease” demonstrate the strategy to develop potent vaccine candidates against Lumpy skin disease. Overall, this manuscript is noteworthy and requires revision to justify its publication in the biomedicines as follows:

Comments

1. Abstract, please add a statement about the novelty and significance of this study.

2. Abstract “Furthermore, according to computer-based analysis, the constructed vaccine provides adequate ……..monkeypox virus.” This sentence can rephrase for broader scenarios, not for a single application.

3. Introduction (As the author mentions monkeypox in the abstract), the authors can add one paragraph on the recent outbreaks of viral infections (Lumpy virus along with a few others i.e. monkeypox and SARS-CoV-2), their mechanism and mortality rates, and the significant challenges/limitations due to their constant evolution via mutations in the development of global vaccines i.e. Computers in Biology and Medicine 153 (2023) 106497; Infection 50 (2022) 309-325.

4. Please provide a brief illustration of the Lumpey virus infection, the mechanism of skin disease, its evolution mechanisms, and prevention strategies.

5. Discussion section can be more elaborated (minor).

6. Please add a “Conclusion” section to highlight the significance of the finding, limitations, and perspectives.

7. Figures quality can be significantly improved in terms of font size and resolutions, etc.

Author Response

Reviewer #4

Comments and Suggestions for Authors

The manuscript by Shahab M et al. entitled “An Immunoinformatics approach to design Novel and potent Multi-Epitope based vaccine to target Lumpy skin disease” demonstrate the strategy to develop potent vaccine candidates against Lumpy skin disease. Overall, this manuscript is noteworthy and requires revision to justify its publication in the biomedicines as follows:

Comments

  1. Abstract, please add a statement about the novelty and significance of this study.

Response: I have made the changes according to the reviewer's comment.

  1. Abstract “Furthermore, according to computer-based analysis, the constructed vaccine provides adequate ……..monkeypox virus.” This sentence can rephrase for broader scenarios, not for a single application.

Response: The sentence has now been rephrased for broader application.

  1. Introduction (As the author mentions monkeypox in the abstract), the authors can add one paragraph on the recent outbreaks of viral infections (Lumpy virus along with a few others i.e. monkeypox and SARS-CoV-2), their mechanism and mortality rates, and the significant challenges/limitations due to their constant evolution via mutations in the development of global vaccines i.e. Computers in Biology and Medicine 153 (2023) 106497; Infection 50 (2022) 309-325.

Response: Modification is made on the basis of reviewer comments.

  1. Please provide a brief illustration of the Lumpy virus infection, the mechanism of skin disease, its evolution mechanisms, and prevention strategies.

Response: I done it according to reviewer comments

  1. Discussion section can be more elaborated (minor).

Response: Done it according to the reviewer comments.

  1. Please add a “Conclusion” section to highlight the significance of the finding, limitations, and perspectives.

Response: Thank you so much for your valuable suggestion according to the comment I have added the conclusion section.

  1. Figures quality can be significantly improved in terms of font size and resolutions, etc.

Response: Done it according to the reviewer comment.

Round 2

Reviewer 2 Report

I'm disappointed with the author's response. A chance given means authors must improve the manuscript, failed to do so leads to rejection. 

".....immunoinformatics based multi-epitope vaccine approach is an attractive alternative."  

Fine, but how can we confirm its efficacy? 

Author Response

Dear reviewer! please find the attachment.

Please consider the Latest_Round 2 response

Reviewer #2

".....immunoinformatics based multi-epitope vaccine approach is an attractive alternative."  

Fine, but how can we confirm its efficacy? 

Response: Thank you so much for your valuable comment! Dear reviewer this work is specifically related to bioinformatics and same type of study already has been published in other journal using bioinformatics tools i.e. (Design of a novel multiple epitope-based vaccine: an immunoinformatics approach to combat monkeypox) in the Journal of Biomolecular Structure and Dynamics, DOI: 10.1080/07391102.2022.2141887, and (Multiepitope-Based Subunit Vaccine Design and Evaluation against Respiratory Syncytial Virus Using Reverse Vaccinology Approach) in the journal of vaccines, DOI:10.3390/vaccines8020288. Employing in silico tools to design means by which to treat specific pathogens enables researchers to modify procedures with little expense according to their objectives. Instances of emergent health crises demand efficiency in drug development, but the expenses associated with vaccine design pose considerable obstacles. While the availability of computational tools has mitigated these challenges, their lack of transparency complicates their practical use.

Reviewer 4 Report

The manuscript should be properly revised as per previous comments.

Additional Notes: 

Please provide detailed point-to-point changes in response.

In addition, authors also need to verify (1) many citations are miss-matched (introduction), (2) improve the quality of many figures.

Author Response

Dear reviewer! please find the attachment.

Please consider Latest_Round 2 response 

Reviewer Comments:

Reviewer #4

The manuscript should be properly revised as per previous comments.

Additional

Comments and Suggestions for Authors

The manuscript by Shahab M et al. entitled “An Immunoinformatics approach to design Novel and potent Multi-Epitope based vaccine to target Lumpy skin disease” demonstrate the strategy to develop potent vaccine candidates against Lumpy skin disease. Overall, this manuscript is noteworthy and requires revision to justify its publication in the biomedicines as follows:

Comments

  1. Abstract, please add a statement about the novelty and significance of this study.

Response:

Despite the availability of information, there is still no competitive vaccine available for LSD. Therefore, the current study was conducted an epitope-based Lumpy skin vaccine that is efficient, secure, and biocompatible that stimulates both innate and adaptive immune responses using Immunoinformatics techniques. Initially, putative virion core proteins were manipulated, B-cell and T-cell epitopes have been predicted and connected with the help of adjuvants and linkers.

  1. Abstract “Furthermore, according to computer-based analysis, the constructed vaccine provides adequate ……..monkeypox virus.” This sentence can rephrase for broader scenarios, not for a single application.

Response: Additionally, computer-based research shows that the constructed vaccine provides adequate population coverage, making it a promising candidate for use in the designing of vaccine against other viruses within the Poxviridae family and potentially other viruses families as well.

  1. Introduction (As the author mentions monkeypox in the abstract), the authors can add one paragraph on the recent outbreaks of viral infections (Lumpy virus along with a few others i.e. monkeypox and SARS-CoV-2), their mechanism and mortality rates, and the significant challenges/limitations due to their constant evolution via mutations in the development of global vaccines i.e. Computers in Biology and Medicine 153 (2023) 106497; Infection 50 (2022) 309-325.

Response:

Monkeypox

To combat human diseases caused by monkey pox virus, close family member of the variola virus, the causative agent of smallpox, which killed 300 million people worldwide in the twentieth century.

SARS-CoV-2

a global outbreak of a respiratory illness [Coronavirus disease (COVID-19)] caused by newly discovered coronavirus variants, SARS-CoV-2, threatened human existence by claiming 54.76 lakhs live until January 5, 2022.

  1. Please provide a brief illustration of the Lumpy virus infection, the mechanism of skin disease, its evolution mechanisms, and prevention strategies.

Response: Lumpy skin disease is caused by the lumpy skin virus (LSDV) which can induce cattle with high fever and extensive nodules on the mucosa or the scarf skin. It is transmitted by blood-feeding insects, such as certain species of flies and mosquitoes, or ticks. Indeed, an efficient vaccine’s effect is its ability to provide life-long immunity, protecting against repeated infection episodes

  1. Discussion section can be more elaborated (minor).

Response: Done it according to the reviewer comments.

  1. Please add a “Conclusion” section to highlight the significance of the finding, limitations, and perspectives.

Response: 

Conclusion

The high burden of Lumpy Skin Disease (LSD), which is associated with the Poxviridae family, is a serious threat to cattle stockbreeding. As such, there is no effective vaccine available for the treatment of LSDV infections. Many antiviral medications have been studied but none have clearly demonstrated effective results against the infection. Reverse vaccinology and computational techniques have been used to build a multi-epitope based subunit vaccine that could activate humoral and cellular immune responses. This study was undertaken to design Multi-epitope based vaccine against the lumpy skin disease by utilizing several bioinformatics tools. The constructed Multi-epitope based vaccine, coupled with computational analysis such as molecular dynamic simulation, C-immune simulation, codon adaptation, and in silico cloning validated our design construct as a suitable vaccine candidate.  The findings provide the way for the construction of the lumpy skin vaccine; however further experimental validation is needed to confirm the vaccine’s reliability, effectiveness, and safety of the vaccine constructs and reduce the lumpy skin disease.

  1. In addition, authors also need to verify (1) many citations are miss-matched (introduction), (2) improve the quality of many figures.

Response: I have improved the figure quality and correct the citation according to the reviewer comment.

Round 3

Reviewer 2 Report

In the future, authors must consider doing some in vitro work. Docking and simulation are not enough to claim efficacies unless proved in the wet lab.  

Reviewer 4 Report

accept